# Comment on Aji et al. aMMP-8 POCT vs. Other Potential Biomarkers in Chair-Side Diagnostics and Treatment Monitoring of Severe Periodontitis. *Int. J. Mol. Sci.* 2024, *25*, 9421

**DOI:** 10.3390/ijms26189072

**Published:** 2025-09-18

**Authors:** Adam Markaryan, Robert Gellibolian, Craig S. Miller, Jeffrey L. Ebersole, Alpdogan Kantarci

**Affiliations:** 1CellectGen, Inc., Pasadena, CA 91107, USA; robert.gellibolian@cellectgen.com; 2Department of Oral Health Practice, College of Dentistry, University of Kentucky, Lexington, KY 40536, USA; craig.miller@uky.edu; 3Department of Biomedical Sciences, School of Dental Medicine, University of Nevada Las Vegas, Las Vegas, NV 89106, USA; jeffrey.ebersole@unlv.edu; 4Department of Immunology, Clinical Research Center, ADA Forsyth Institute, Cambridge, MA 02142, USA; akantarci@forsyth.org

We read with interest the August *International Journal of Molecular Science* publication (Aji, N.R.A.S.; Räisänen, I.T.; Rathnayake, N.; Lundy, F.T.; Mc Crudden, M.T.C.; Goyal, L.; Sorsa, T.; Gupta, S. aMMP-8 POCT vs. Other Potential Biomarkers in Chair-Side Diagnostics and Treatment Monitoring of Severe Periodontitis. *Int. J. Mol. Sci.* 2024, 25, 9421). Our interest focused on how the authors compared the effectiveness of different potential biomarkers of periodontitis, including active matrix-metalloproteinase-8 (aMMP-8) and total MMP-8, to diagnose and monitor the effects of nonsurgical periodontal therapy. The data presented is compelling, but there are a few concerns and questions that we would like to bring to the attention of the authors and editors.

The first involves the comparison between the aMMP-8 point-of-care test (POCT) and the MMP-8 enzyme activity assay. The question is whether the detection antibody used in the POCT recognizes the active MMP-8 (aMMP-8). A few publications suggest that these particular fragment specific monoclonal antibodies, 1491-E6-F7 and 1492-B3-C11, detect the 20–35 kDa products of MMP-8, not the functionally active enzyme [1,2]. Although it is unknown, these small fragments are unlikely to display the full collagenolytic activity as full-length (52 kDa) aMMP-8. A more direct correlation between the collagenase activity and periodontal tissue destruction has been established by Lee et al. [3], leading us to believe that a functionally active MMP-8 (52 kDa) is required for tissue destruction.

Second, the authors referenced using an aMMP-8-specific monoclonal antibody from Merck Millipore that selectively captures aMMP-8. This is a non-specific antibody, recognizing both the active and latent (inactive) forms of the MMP-8 enzyme. This lack of specificity likely affects the interpretation of measured aMMP-8 levels, raising questions regarding what is actually being measured in the assays.

Third, the FRET substrate [4] used to measure MMP-8 enzyme activity lacks the necessary conserved Gly–Leu sequence, which is a required cleavage site in collagen [5]. Again, this brings into question whether it is collagen breakdown that is being measured or a non-specific target of MMPs in general.

Fourth, assuming that the POCT is specific for detecting aMMP-8 and the authors have access to this antibody, why not use it to perform the capture assay? Why use Merck Millipore’s antibody, which as mentioned above, lacks the ability to differentiate between the active and latent forms of MMP-8?

We appreciate the response of Aji et al. 2025 to the four questions in our comment, which were primarily focused on gaining clarity on crucial methodological details of active (aMMP-8) POCT and activity assays in the original article [6]. However, the authors’ reply appears as mainly a comprehensive review of the literature to justify the validity and robustness of the aMMP-8 POCT assay, as well as its cut-off values. Very little detail was offered with respect to the specifics of the four points of concern that we originally raised.

For example, our first question was whether the detection mAb used in the aMMP-8 POCT specifically recognizes active MMP-8? According to the authors’ reply, the mAb that was described by Sorsa et al.’s US patent 2019 [2] is designed to detect “*active MMP8, its activation products and related lower MW fragments*”. However, the Western blots provided (Figure 1A,B) show that this mAb mostly detects pro-MMP-8 and 20–30 kDa fragments, with traces of aMMP-8. Aside from difficulty seeing differences between healthy (lanes 1–6) and periodontitis (lanes 7–12) samples in Figure 1, it is unclear how, under the denaturing conditions of the Western blot, the mAb would be able to differentiate between the pro-MMP-8 and aMMP-8 forms. Thus, the question we posed regarding the specificity of this mAb in binding the active and not pro-MMP8 still remains unanswered.

The second, third, and fourth questions are focused on understanding the details of the aMMP-8 assay using the FRET peptide substrate [4]. This allowed the authors to establish a correlation between the aMMP-8 POCT and an MMP-8 enzymatic activity assays. In the original article [6], Aji et al. described using an aMMP-8-specific antibody from Merck to coat the wells of a plate for selectively capturing aMMP-8 before the assay. As mentioned in our first response, the Merck antibody non-specifically binds both active and latent forms of the MMP-8 enzyme. Given this fact, our question was specifically about why use the non-specific Merck antibody at all? Why not use the specific aMMP8 mAb that the authors developed and have at their disposal? The authors’ mentioned in reply that their overall goal was to compare the POCT with an “*entirely independent catalytic assay*”, which would exclude the use of the same antibody in both methods. Since the specificity of the assay is defined by the capture antibody, and not by the FRET substrate, this raises questions about the potential blocking effects of latent MMP-8 on this binding and its impact on the accuracy of aMMP-8 quantitation. This concern is not addressed in the original article nor adequately addressed in the authors’ reply. Additionally, the composition of the FRET substrate lacks the conserved “Gly–Leu sequence”, which is a cleavage site in collagen molecules for aMMP-8. Thus, future studies are likely needed using a more natural fluorogenic collagen substrate to assure the measurement of aMMP-8 in these types of assays.

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
