# Peer review of "Comment on Aji et al. aMMP-8 POCT vs. Other Potential Biomarkers in Chair-Side Diagnostics and Treatment Monitoring of Severe Periodontitis. Int. J. Mol. Sci. 2024, 25, 9421"

_ijms, 2025, doi:10.3390/ijms26189072_

Round 1

Reviewer 1 Report

Comments and Suggestions for Authors

Dear authors,

Agree with all 4 comments. Further clarification will contribute to better understanding of the interplay between active matrix-metalloproteinase-8 (aMMP-8), total MMP-8 specific (or less specific) POC diagnostic platforms

Reviewer 2 Report

Comments and Suggestions for Authors

1.You mention the use of specific monoclonal antibodies (1491-E6-F7 and 1492-B3-C11) for detecting aMMP-8. However, current evidence suggests these antibodies may detect degradation fragments (20–35 kDa) rather than the full-length active MMP-8 enzyme (52 kDa). Please clarify whether the POCT method used in your study reliably detects functionally active MMP-8 and provide supporting validation data or references.

2. Your study also refers to the use of a commercial antibody from Merck Millipore, which cannot reportedly distinguish between active and latent forms of MMP-8. Since your conclusions depend on measuring active MMP-8, it is essential to explain why you selected this antibody, and to discuss how this may affect the interpretation of your data.

3. If your team has access to a POCT antibody specific to active MMP-8, it is unclear why this was not used in the laboratory-based capture assay. Please clarify the reasoning behind using different antibodies across assays and discuss whether this could introduce discrepancies between methods.

4. Please elaborate on whether your diagnostic approach aims to identify total MMP-8 protein presence or actual enzymatic activity. This distinction is crucial, particularly when positioning MMP-8 as a biomarker for tissue destruction and treatment monitoring.

I encourage you to revise the manuscript with detailed clarifications and justifications addressing the points above. Strengthening the methodological transparency will enhance the reliability and clinical applicability of your findings.

Reviewer 3 Report

Comments and Suggestions for Authors

This is a comment on a previously published paper entitled aMMP-8 POCT vs. Other Potential Biomarkers in Chair-Side Diagnostics and Treatment Monitoring of Severe Periodontitis. 

The authors pointed out scientifically grounded points regarding the methodology adopted by Aji et al., which may contribute to future studies.
